# Synthesis and Structure of a Ferrocenylsilane-Bridged Bisphosphine †

**Takahiro Sasamori** [1,2,*] , **Hiromu Ueno** [3] **and Shogo Morisako** [1,2]

1    Division of Chemistry, Faculty of Pure and Applied Sciences, University of Tsukuba, 1-1-1 Tennoudai, Ibaraki, Tsukuba 305-8571, Japan; morisako.shogo.gf@u.tsukuba.ac.jp

2    Tsukuba Research Center for Energy Materials Sciences (TREMS), University of Tsukuba, 1-1-1 Tennoudai, Ibaraki, Tsukuba 305-8571, Japan

3    College of Chemistry, School of Science and Engineering, University of Tsukuba, 1-1-1 Tennoudai, Ibaraki, Tsukuba 305-8571, Japan; u.hiromu@dmb.chem.tsukuba.ac.jp

\*    Correspondence: sasamori@chem.tsukuba.ac.jp; Tel.: +81-29-853-4412

†    Dedicated to Prof. Dr. Norihiro Tokitoh on the occasion of his 65th birthday.

**Abstract:** A bisferrocenylsilane-bridged bisphosphine, i.e., a bisphosphine bridged by bis(1′-dicyclohexylphosphino-1-ferrocenyl)dimethoxysilane, was synthesized and structurally characterized. Its redox behavior was examined by cyclic voltammetry and differential pulse voltammetry, which revealed two-step oxidation processes.

**Keywords:** ferrocenyl phosphine; bisphosphine; silane; redox behavior; cyclic voltammetry

## 1. Introduction

Organic bisphosphines, i.e., organic molecules that bear two phosphine moieties as coordinating sites, represent an important class of ancillary ligands in transition-metal coordination chemistry and catalysis [1–4]. Especially transition-metal-based catalysts in synthetic organic chemistry often require specific custom-tailored bisphosphine ligands in order to realize their full catalytic performance [1–5]. Furthermore, bisphosphine ligands can be applied as bridging blocks in metal-organic-frameworks (MOFs) [6–9]. In this context, bisphosphine ligands that contain a redox-active moiety would be of great interest, as such redox-active bisphosphine ligands could potentially be functionalized, which would possibly afford control over the chemical/physical properties of the resulting electrochemical-stimulus-responsive transition-metal catalysts and MOFs. Given that the ferrocenyl group is a redox-active framework that can be easily modified by a variety of well-established organic synthetic methods [10,11], ferrocenyl bisphosphines could potentially serve as appropriate models for redox-controllable bisphosphine ligands [12]. Bisphosphametallocenes are potentially redox-active bisphosphine ligands given the characteristic redox behavior of the metallocene framework, and the isolation of several bisphosphametallocene derivatives has already been reported [13]. However, in most cases, these phosphametallocenes have only been used as simple organic bisphosphines that can chelate onto a transition metal with a relatively rigid structure based on the h$^5$-sandwich-type skeleton, which often conceals their redox behavior. Furthermore, a ferrocenylsilane polymer has attracted much attention as an optoelectronic material due to the redox behavior of both oligosilanes and ferrocenyl moieties [14–19]. Based on the combined consideration of the functions of both redox-active bisphosphine ligands and ferrocenylsilanes, we have designed a novel type of a redox-active bisphosphine, the two phosphine moieties of which are bridged by a bis(ferrocenyl)silene linker (Figure 1). The silyl linkage should enable electronic communication between the two 1′-biscyclohexylphosphino-1-ferrocenyl groups, and work as the redox-active bisphosphine ligand. In addition, the alkoxy groups on the silicon atom, which is bridging the redox-active moieties, should be an appropriate model

for a supporting moiety on silicone polymers. Here, we report the synthesis and solid-state structure of the ferrocene-based bisphosphine ligand together with its redox behavior, which was examined by cyclic voltammetry and differential pulse voltammetry.

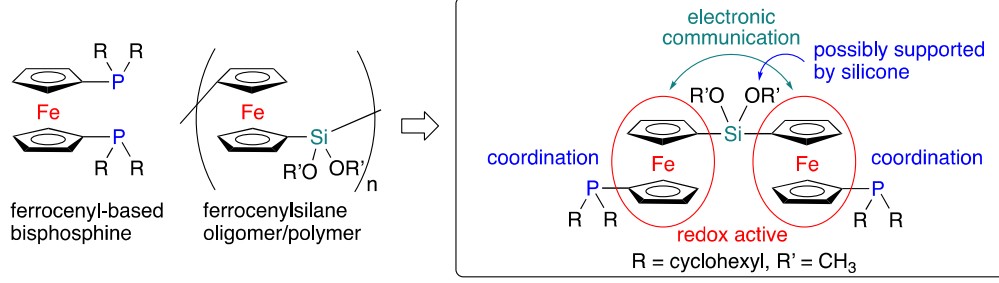

**Figure 1.** Molecular design of redox-active bisphosphine ligand.

## 2. Results and Discussion

According to literature procedures, 1-bromo-1′-dicyclohexylphosphinoferrocene was prepared [20,21]. 1-Lithio-1′-dicyclohexylphosphinoferrocene (Li-fc-PCy$_2$) was prepared by the reaction of 1-bromo-1′-dicyclohexylphosphinoferrocene with *n*-BuLi in THF at −60 °C. When an excess of Si(OMe)$_4$ was added to the THF solution of Li-fc-PCy$_2$ at −60 °C, the corresponding ferrocenyltrimethoxysilane (**1**) was isolated in 42% yield (Scheme 1). When the THF solution of Li-fc-PCy$_2$ was treated with half an equivalent of Si(OMe)$_4$ under otherwise identical conditions, bis(ferrocenyl)silane **2** was obtained in 46% isolated yield. Thus, the number(s) of the introduced ferrocenyl ligand(s) can be controlled by the equivalent(s) of Si(OMe)$_4$ relative to 1-lithio-1′-dicyclohexylphosphinoferrocene. The obtained ferrocenylphosphine ligands **1** and **2** were identified by multinuclear NMR spectroscopy and mass spectrometry as well as structurally characterized by single-crystal X-ray diffraction analysis.

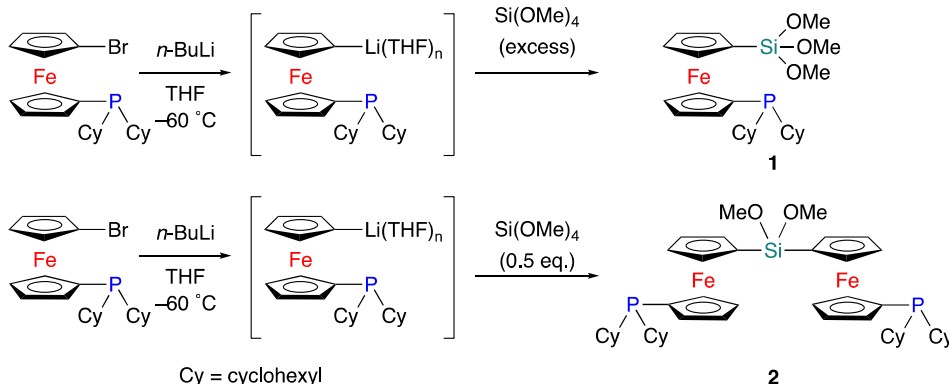

**Scheme 1.** Synthesis of (1′-phosphino-1-ferrocenyl)trimethoxysilane **1** and bis(1′-phosphino-1-ferrocenyl)dimethoxysilane **2**.

In the $^{29}$Si and $^{31}$P NMR spectra of **1** and **2**, *J*-coupling between the nuclei was not observed, suggesting negligible P→Si coordination in solution. Thus, it can be expected that the phosphine moieties in **1** and **2** may serve as intermolecular coordinating sites. In the crystalline state, the molecular structures of (1′-phosphino-1-ferrocenyl)trimethoxysilane **1** and bis(1′-phosphino-1-ferrocenyl)dimethoxysilane **2** show that the phosphine moieties are, as in solution, free from intramolecular coordination (Figure 2). The unit cell of **1** contains two crystallographically independent molecules, which differ with respect to the orientation of the methoxy groups. The observed structural parameters of **1** and **2** are comparable, suggesting that the structural/electronic perturbation in the bisphosphine framework of **2** relative to that of monophosphine **1** should be negligible with regard to the

coordinating ability. The P–C(Fc) and P–C(Cy) bond lengths of **1** are 1.8237(13)/1.8214(13) Å and 1.8594(13)/1.8652(13)/1.8580(13)/1.8667(13) Å, respectively, while those of **2** are 1.823(3)/1.825(3) Å and 1.863(3)/1.871(3)/1.860(3)/1.877(3), respectively. The P–C(Fc) bonds, which are slightly shorter than the P–C(Cy) bonds in both **1** and **2**, should most likely be interpreted in terms of the sp$^2$ character of the ferrocenyl carbon atoms and a possible hyperconjugation between the π-orbitals of the ferrocenyl moieties and the σ*(P–C(Cy)) orbitals. Moreover, the high values of the sum of the C–P–C angles in both **1** (ca. 305°) and **2** (ca. 303°) suggest a strong σ-coordinating ability for the P atoms.

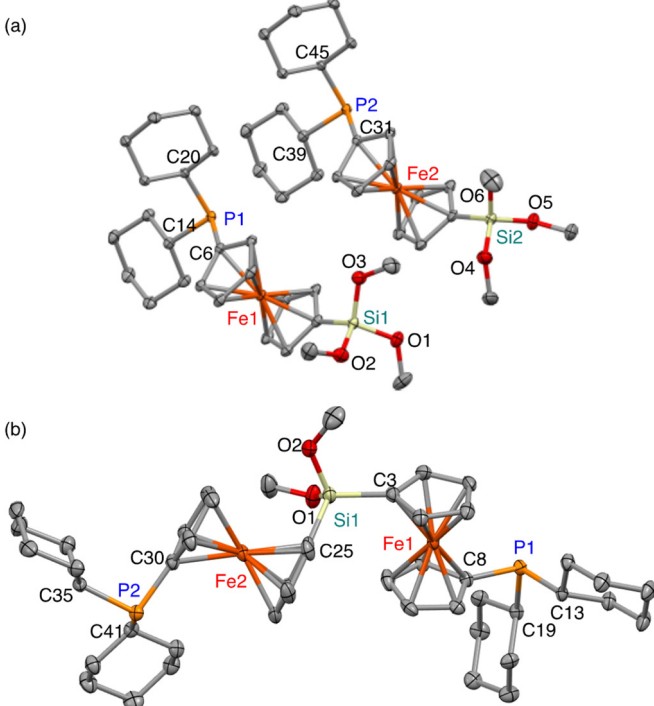

**Figure 2.** Molecular structures of (**a**) (1′-phosphino-1-ferrocenyl)trimethoxysilane **1** and (**b**) bis(1′-phosphino-1-ferrocenyl)dimethoxysilane **2** with thermal ellipsoids at 50% probability. Selected bond lengths (Å) and angles (○): (**a**) **1**, P1–C6, 1.8237(13), P1–C14, 1.8594(13), P1–C20, 1.8652(13), P2–C31, 1.8214(13), P2–C39, 1.8580(13), P2–C45, 1.8667(13), Si1–C1, 1.8331(14), Si1–O1, 1.6295(11), Si1–O2, 1.6215(11), Si1–O3, 1.6171(11), Si2–C(26), 1.8313(14), Si2–O4, 1.6212(11), Si2–O5, 1.6259(11), Si2–O6, 1.6193(11), C6–P1–C14, 102.15(6), C6–P1–C20, 101.59(6), C14–P1–C20, 102.04(6), C31–P2–C39, 101.95(6), C31–P2–C45, 101.07(6), C35–P2–C45, 101.98(6). (**b**) **2**, P1–C8, 1.823(3), P1–C13, 1.863(3), P1–C19, 1.871(3), P2–C30, 1.825(3), P2–C35, 1.877(3), P2–C41, 1.860(3), Si1–C3, 1.841(3), Si1–C25, 1.840(3), Si1–O1, 1.627(3), Si1–O2, 1.637(3), C3–Si1–C25, 110.11(14), C8–P1–C13, 100.71(13), C8–P1–C19, 100.00(13), C13–P1–C19, 102.07(13), C30–P2–C35, 98.92(14), C30–P2–C41, 102.54(14), C35–P2–C41, 101.87(14).

The theoretically optimized structural parameters for **1** and **2**, calculated at the B3PW91-D3(BJ)/6-311G(3d) level of theory, were in good agreement with those obtained from the XRD analyses (See, Supporting Information.). For example, the optimized P–C(Fc)/P–C(Cy) bonds of **1** and **2** are 1.82/1.86 Å in both cases, while the high values of the sum of the C–P–C angles in **1** and **2** are ca. 302°. Thus, further theoretical calculations were carried out at the same level of theory [22].

The redox behavior of bisphosphine **2** was examined using cyclic voltammetry (CV) and differential pulse voltammetry (DPV). The cyclic voltammogram of **2** showed a pseudo-reversible two-step redox wave in the oxidation region (Figure 3a). The oxidation potentials of **2** ($E^{OX}_1$ = 0.20 V; $E^{OX}_2$ = 0.43 V) were determined based on both CV and DPV, even though the $E_{pa1}$ peak for the first oxidation process and the $E_{pc2}$ peak for the reduction process of the dicationic species were not observed clearly in the cyclic voltammogram.

Monophosphine **1** showed a one-step oxidation wave ($E_{1/2}$ = 0.30 V) at a higher potential relative to those of **2** (Figure 3b), suggesting an extended conjugation between the two ferrocenyl moieties in **2** via σ-conjugation with the Si(OMe)$_2$ linker. The separation of the oxidation potentials for **2** (ΔE = 0.23 V; at –30 °C) can be converted into the corresponding comproportionation constant $K_{com}$ for the equilibrium **2²⁺** + **2** ⇌ 2 **2⁺** ($K_{com}$ = 5.9 × 10⁴) [23], which suggests strong electronic communication between the two 1′-phosphinoferrocenyl moieties in **5** under oxidative conditions.

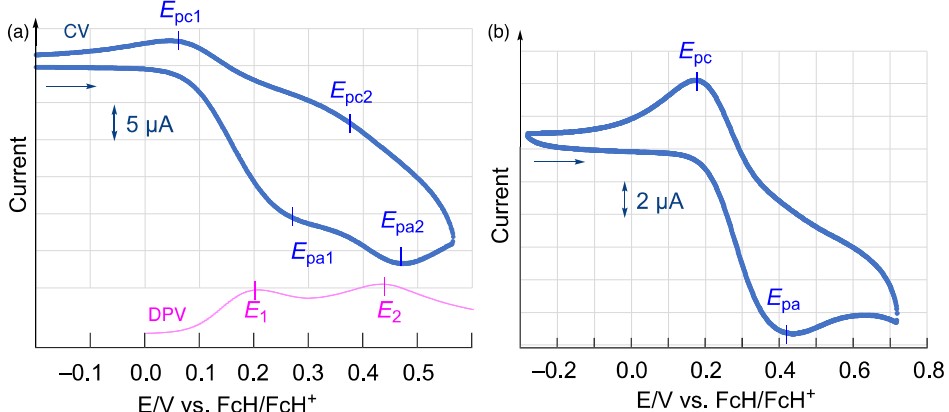

**Figure 3.** (**a**) Cyclic (blue) and differential-pulse (red) voltammograms of bisphosphine **2**. (**b**) Cyclic voltammogram of monophosphine **1**; measured in CH$_2$Cl$_2$ at –30 °C (1.0 mM; supporting electrolyte: 0.1 M [$n$Bu$_4$N][PF$_6$]; scan rate: 100 mVs⁻¹).

The theoretically calculated adiabatic ionization energies for **1** (6.00 eV) and **2** (5.89 eV) suggest a lower oxidation potential for **2** relative to that of **1** [22]. As shown in Figure 4, the calculated spin density for **2⁺** is delocalized around the ferrocenyl moiety; in contrast, that of **1⁺** is localized on the iron atom. Furthermore, the triplet state of **2²⁺** was found to be by 25 kcal/mol more stable than the singlet state of **2²⁺**, whereby the spin density of **2²⁺** was predominantly spread on one of the ferrocenyl moieties together with small contributions from the other ferrocenyl moiety and the phosphine atom.

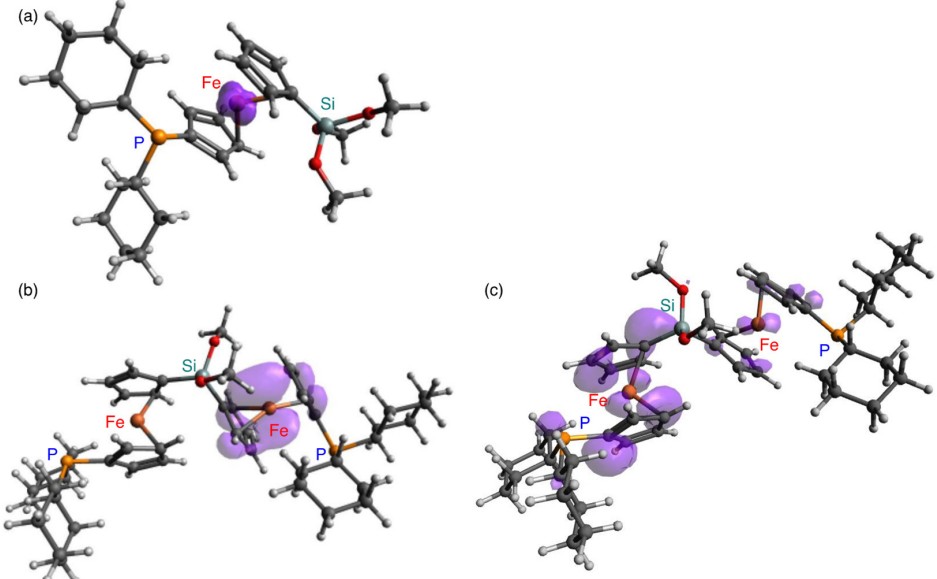

**Figure 4.** Theoretical spin densities (purple) of (**a**) radical cation **1⁺**, (**b**) radical cation **2⁺**, and (**c**) dication **2²⁺** (triplet state), calculated at the B3PW91-D3(BJ)/6-311G(3d) level of theory.

## 3. Conclusions

Silyl-tethered bis(ferrocenylphosphine) **2** was successfully synthesized and structurally characterized. The redox behavior of bisphosphine **2** and monophosphine **1** was examined based on cyclic voltammetry and differential pulse voltammetry. Bisphosphine **2** is oxidized at a lower oxidation potential compared to that of monophosphine **1** due to the electronic communication between the two ferrocenylphosphine moieties through the silyl linker. Further investigations into the complexation of **5** with transition metals and the creation of redox-active MOF systems that contain **2** are currently in progress in our laboratory.

## 4. Materials and Methods

### 4.1. General Information

All manipulations were carried out under an argon atmosphere using Schlenk-line techniques. $^{1}$H, $^{13}$C, $^{29}$Si, and $^{31}$P NMR spectra were measured on a Bruker AVANCE-400 spectrometer ($^{1}$H: 400 MHz; $^{13}$C: 101 MHz; $^{29}$Si: 79.5 MHz, $^{31}$P: 162 MHz). Signals arising from residual $C_6D_5H$ (7.16 ppm) in $C_6D_6$ were used as the internal standard for the $^{1}$H NMR spectra, that of $C_6D_6$ (128.0 ppm) was used for the $^{13}$C NMR spectra, and external $SiMe_4$ (0.0 ppm) and external 85% $H_3PO_4$ in a capillary (0.0 ppm) were used for the $^{29}$Si and $^{31}$P NMR spectra, respectively. High-resolution mass spectra were obtained on a JEOL JMS-T100CS (APCI) mass spectrometer. All melting points were determined on a Büchi Melting Point Apparatus M-565 and are uncorrected. 1-Bromo-1′-dicyclohexylphosphinoferrocene was prepared according to a reported procedure [20,21].

### 4.2. Synthesis of Monophosphine 1

Fc$^P$Br (122 mg, 0.264 mmol) was dissolved in THF (0.40 mL) and cooled to −60 °C. After a hexane solution of *n*-BuLi (0.100 mL, 2.64 M in hexane, 0.264 mmol) had been added dropwise, the solution was kept at −60 °C for 3.0 h. Then, $Si(OMe)_4$ (0.195 mL, 1.32 mmol) was added at –78 °C. After stirring the reaction mixture for 18 h at room temperature, all volatiles were removed under reduced pressure and hexane was added. After all insoluble inorganic salts had been removed by filtration through a pad of celite, the solvent was removed from the filtrate under reduced pressure to give an orange solid that was purified by column chromatography ($SiO_2$, hexane:ethyl acetate = 10:1 (*v/v*)). Removal of the solvent of the obtained fractional solution gave monophosphine **1** (55.5 mg, 0.110 mmol, 42%) as orange crystals. Mp. 76–82 °C. $^{1}$H NMR (400 MHz, $C_6D_6$) δ 1.05–1.41 (m, 10H), 1.56–1.93 (m, 10H), 1.99–2.10 (m, 2H), 3.58 (s, 9H), 4.29 (AA′BB′C system, pseudo-q, *J* = 1.6 Hz, 2H), 4.33–4.37 (m, 4H), 4.44 (AA′BB′ system, pseudo-t, *J* = 1.8 Hz, 2H); $^{13}$C{$^{1}$H} NMR (101 MHz, $C_6D_6$) δ 26.9 (CH$_2$), 27.6 (d, $J_{CP}$ = 8.1 Hz, CH$_2$), 27.7 (d, $J_{CP}$ = 11 Hz, CH$_2$), 30.6 (d, $J_{CP}$ = 11 Hz, CH$_2$), 30.7 (d, $J_{CP}$ = 14 Hz, CH$_2$), 34.0 (d, $J_{CP}$ = 14 Hz, CH), 50.7 (CH$_3$), 61.7 (C), 70.8 (d, $J_{CP}$ = 2.0 Hz, CH), 72.4 (d, $J_{CP}$ = 11 Hz, CH), 73.6 (d, $J_{CP}$ = 2.0 Hz, CH), 75.0 (CH), 78.0 (d, $J_{CP}$ = 21 Hz, C); $^{29}$Si{$^{1}$H} NMR (79.5 MHz, $C_6D_6$) δ −47.2; $^{31}$P{$^{1}$H} NMR (162 MHz, $C_6D_6$) δ −8.2; HRMS (APCI), *m/z*: Found: 503.18039 ([M+H]$^+$), calculated for $C_{25}H_{40}FeO_3PSi$ ([M+H]$^+$): 503.18341.

### 4.3. Synthesis of Bisphosphine 2

Fc$^P$Br (122 mg, 0.264 mmol) was dissolved in THF (0.40 mL) and cooled to −60 °C. After a hexane solution of *n*-BuLi (0.100 mL, 2.64 M in hexane, 0.264 mmol) had been added dropwise, the solution was kept at −60 °C for 3.0 h. Then, $Si(OMe)_4$ (0.020 mL, 0.132 mmol) was added at –78 °C. After stirring the reaction mixture for 15 h at room temperature, all volatiles were removed under reduced pressure and hexane was added. After all insoluble inorganic salts had been removed by filtration through a pad of celite, the solvent was removed from the filtrate under reduced pressure to give an orange oil that was subjected to column chromatography ($SiO_2$, hexane:ethyl acetate = 20:1 (*v/v*)). Storage of a hexane solution of the obtained orange oil at −30 °C gave an orange suspension. After centrifugal separation and decantation, the supernatant was removed by using a syringe. Washing the residual solid with hexane gave bisphosphine **2** (52.1 mg, 0.061 mmol, 46%) as orange

crystals. Mp. 96–106 °C. $^1$H NMR (400 MHz, $C_6D_6$) δ 1.06–1.42 (m, 20H), 1.56–1.94 (m, 20H), 2.01–2.12 (m, 4H), 3.69 (s, 6H), 4.26 (AA′BB′C system, pseudo-q, *J* = 1.6 Hz, 4H), 4.38–4.45 (m, 12H); $^{13}$C{$^1$H} NMR (101 MHz, $C_6D_6$) δ 26.9 ($CH_2$), 27.6 (d, $J_{CP}$ = 9.1 Hz, $CH_2$), 27.8 (d, $J_{CP}$ = 11 Hz, $CH_2$), 30.6 (d, $J_{CP}$ = 10 Hz, $CH_2$), 30.7 (d, $J_{CP}$ = 14 Hz, $CH_2$), 34.0 (d, $J_{CP}$ = 13 Hz, CH), 51.1 ($CH_3$), 66.3 (C), 70.9 (d, $J_{CP}$ = 2.0 Hz, CH), 72.4 (d, $J_{CP}$ = 10 Hz, CH), 73.6 (d, $J_{CP}$ = 2.0 Hz, CH), 74.9 (CH), 77.9 (d, $J_{CP}$ = 21 Hz, C); $^{29}$Si{$^1$H} NMR (79.5 MHz, $C_6D_6$) δ −17.3; $^{31}$P{$^1$H} NMR (162 MHz, $C_6D_6$) δ −8.2; HRMS (APCI), *m/z*: Found: 853.31297 ([M+H]$^+$), calculated for $C_{46}H_{67}Fe_2O_2P_2Si$ ([M+H]$^+$): 853.30867.

### 4.4. X-ray Crystallographic Analysis of *1* and *2*

Single crystals of **1** and **2** were obtained after recrystallization from hexane. Intensity data for **1** and **2** were collected on a Bruker APEX-II system using Mo-Kα radiation (λ = 0.71073 Å), while the preliminary data were collected on the BL02B1 beamline of SPring-8 (proposal numbers: 2020A0557, 2020A1056, 2020A1644, 2020A1650, 2020A0834, 2021A1592, 2021A1578, 2021B1435, and 2021B1833) on a PILATUS3 X CdTe 1M camera using synchrotron radiation (λ = 0.4148 Å). The structures were solved using SHELXT-2018 and refined by a full-matrix least-squares method on $F^2$ for all reflections using SHELXL-2018 [24]. All non-hydrogen atoms were refined anisotropically, and the positions of all hydrogen atoms were calculated geometrically and refined as riding models. Supplementary crystallographic data were deposited at the Cambridge Crystallographic Data Centre (CCDC) under deposition numbers CCDC-2141476 (**1**) and CCDC-2141477 (**2**); these can be obtained free of charge via www.ccdc.cam.ac.uk/data_request.cif (accessed on 29 January 2022).

### 4.5. Electrochemical Measurements

Cyclic and differential-pulse voltammograms were recorded on an ALS 1140A potentiostat/galvanostat using Pt wire electrodes under an argon atmosphere in custom-tailored glassware. Voltammograms were recorded at −30 °C on $CH_2Cl_2$ solutions ((analyte): 1.0 mM; supporting electrolyte: 0.1 M [$n$Bu$_4$N][PF$_6$]) using a variety of scan rates.

### 4.6. Theoretical Calculations

Theoretical calculations for the geometry optimization and frequency calculations of **1**, **2**, **1$^+$**, **2$^+$**, **2$^{2+}$(singlet)**, and **2$^{2+}$(triplet)** were carried out using the Gaussian 16 (Revision C.01) program package [22]. Geometry optimizations were performed at the B3PW91-D3(BJ) level of theory using the 6-311G(3d) basis sets. Minimum energies for the optimized structures were confirmed by frequency calculations. Computational time was generously provided by the Supercomputer Laboratory at the Institute for Chemical Research (Kyoto University). The coordinates of the optimized structures are included in the corresponding .xyz files as supporting information.

**Supplementary Materials:** The following supporting information can be downloaded at: https://www.mdpi.com/article/10.3390/inorganics10020022/s1, Theoretically optimized coordinates (xyz) are available in the Supplementary Materials.

**Author Contributions:** Conceptualization: T.S.; resources and funding acquisition: S.M. and T.S.; experiments: H.U. and S.M.; writing—original draft preparation: T.S. and H.U.—review and editing: T.S., H.U. and S.M.; project administration: T.S. All authors have read and agreed to the published version of the manuscript.

**Funding:** This research was funded by JSPS KAKENHI grants 19H02705 and 21K14607, as well as by 21KK0094 from MEXT (Japan), by the University of Tsukuba Basic Research Support Program Type B, by the Collaborative Research Program of the Institute for Chemical Research at Kyoto University (2018–2022), by the TOBE MAKI Scholarship Foundation (20-JA-014), and by JST CREST grant JPMJCR19R4.

**Data Availability Statement:** Data available in a publicly accessible repository.

**Acknowledgments:** We acknowledge the Supercomputer Laboratory in the Institute for Chemical Research of Kyoto University for the resources used. We would like to thank Toshiaki Noda and Hideko Natsume (Nagoya University) for the expert manufacturing of custom-tailored glassware.

**Conflicts of Interest:** The funders had no role in the design of the study; in the collection, analysis, or interpretation of data; in the writing of the manuscript; or in the decision to publish the results.

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
