# Peer review of "Synthesis and Structure of a Ferrocenylsilane-Bridged Bisphosphine †"

_inorganics, doi:10.3390/inorganics10020022_

Round 1
Reviewer 1 Report
The ms entitled ‘’Synthesis and Structure of a Ferrocenylsilane-bridged Bisphosphine’’ describes the synthesis and structural characterization of two novel mono and bisphosphine ligands and the redox properties of these compounds. Phosphines are ligands with a long history in metal coordination chemistry and in catalysis and the modern day homogeneous transition metal catalysis is underpinned by phosphine ligands and metal–phosphine coordination complexes, and thus phosphine chemistry is a ‘’hot topic’’. This ms is well written and the authors have well characterized the two structures with various of spectroscopic techniques among with X-ray crystallography. The introduction section is adequate and the results and discussion section describes in detail the crystal structure and the redox properties of these two new compounds. Also, the experimental data accompanied with DFT calculations that supports the redox active character of the compounds. In general, this ms deserves acceptance in journal ‘’Inorganics’’ after minor revision points. The communication-when published-will attract the interest of researchers working in the general areas of inorganic and organometallic chemistry of phosphines.
Specific revision points/comments/suggestions raised from this ms are:
- Introduction section: Please revise the sentence starting ‘’Even…behavior’’ line 30-34, to make more sense.
- In the introduction section you mention that the solid state structure of 1 has been determined. In the Result and discussion section this compound is annotated as 5. There is confusion with the numbering of the compounds. I suggest revising the number of two compounds. For example, in Fig.1 there is no need for numbering. You can only assign the R and R’ groups to Cy and Me that lead to the two new corresponding structures. Also, a change in numbering of the products would be more preferable. You can number only the two new compounds and not the starting materials and the intermediates.
- In the Result and Discussion please revise the meaning of sentence starting from line 53 where you describe the formation of the two compounds. As far as I can understand, first you prepared the Br-Ferrocenyl compound and then the Li-(THF)Ferrocenyl. From this compound the two new mono and bisphoshine compounds were afforded after treatment with Si(OMe).
- It would be clearer to accompany Scheme 1 with chemical equations that led to the two new compounds.
- Line 73: ‘’Crystallographically independent’’ instead of ‘’independent’’
- A more detailed description of the two structures would be preferable.
- The XRD data are missing from the Supplementary Information.
- In Reference section please transfer all crystallographic data (Ref.23) in a Table and move it in Supporting Information.
Author Response
We appreciate the positive comment from the reviewer. Please find attached pdf file including replies to Reviewer 1 .

Reviewer 2 Report
In this work, Sasamori et al. reported the synthesis and structure characterization of a bisferrocenylsilane-bridged bisphosphine. The redox behavior was examined to include a two-step oxidation process. And it was found that the bisphosphine compound could be oxidized at lower oxidation potential compared to that of the monophosphine compound, probably due to the electronic communication between the two ferrocenylphosphine moieties through the silyl linker. The authors also conducted DFT computational studies to reveal the spin density distribution of the phosphine compound. This study represents an example of redox behavior of ferrocenylphosphine compounds. However, only one bisphosphine ligand was synthesized and characterized. The relationship between phosphine compound structure and redox behavior couldn’t be revealed. Moreover, the reviewer found that there are several cases, where the citations are inappropriate or incorrect. Overall, major revision is necessary before this work could be published in inorganics.
- As mentioned above, only one bisphosphine ligand was synthesized and characterized. The authors need to synthesize more bisphosphine compounds with different substituents (such as tert-butyl, phenyl, 3-5-di-tert-butylphenyl) on phosphine atoms to reveal the relationship between redox behavior and phosphine compound structures.
- For phosphine compounds as ancillary ligands in transition-metal-catalysis, the authors gave ref. 1-3, which are very specific and limited examples. Reviews and book chapters regarding phosphine ligands should be cited, instead of ref. 1-3. Moreover, for ref. 12, the reviewer believes it couldn’t sever as an example for redox controllable bisphosphine ligands.
- The authors did DFT computational studies and mentioned that the calculated structural parameters were in good agreement with those obtained from the XRD analysis. The comparison of the structural parameters such as bond lengths and angles need to be present in the SI.
Author Response
We appreciate the comments and suggestions from this reviewer. Please find attached pdf file including replies to Reviewer 2 .

Round 2
Reviewer 1 Report
The ms has been improved significantly since the last time it was submitted . The authors have taken into consideration all the comments and suggestions raised from the first submission and thus I belive that this ms can be published as a communication in Inorganics-MDPI in its current form.
Reviewer 2 Report
The reviewer has reviewed the revised manuscript again and found most issues were addressed. The reviewer would be happy to recommend the revised manuscript for publication in Inorganics.